# What Has Longitudinal ‘Omics’ Studies Taught Us about Irritable Bowel Syndrome? A Systematic Review

**DOI:** 10.3390/metabo13040484

**Published:** 2023-03-28

**Authors:** Qin Xiang Ng, Chun En Yau, Clyve Yu Leon Yaow, Ryan Ian Houe Chong, Nicolette Zy-Yin Chong, Seth En Teoh, Yu Liang Lim, Alex Yu Sen Soh, Wee Khoon Ng, Julian Thumboo

**Affiliations:** 1Health Services Research Unit, Singapore General Hospital, Singapore 169608, Singapore; 2NUS Yong Loo Lin School of Medicine, Singapore 117597, Singapore; 3Department of Gastroenterology and Hepatology, Tan Tock Seng Hospital, Singapore 308433, Singapore; 4Department of Medicine, Alexandra Hospital, National University Health System, 378 Alexandra Road, Singapore 159964, Singapore; 5Department of Rheumatology and Immunology, Singapore General Hospital, Singapore 169608, Singapore; 6SingHealth Duke-NUS Medicine Academic Clinical Programme, Duke-NUS Medical School, Singapore 169857, Singapore

**Keywords:** irritable bowel syndrome, omics, metabolomics, mechanisms, review

## Abstract

Irritable bowel syndrome is a prototypical disorder of the brain–gut–microbiome axis, although the underlying pathogenesis and mechanisms remain incompletely understood. With the recent advances in ‘omics’ technologies, studies have attempted to uncover IBS-specific variations in the host–microbiome profile and function. However, no biomarker has been identified to date. Given the high inter-individual and day-to-day variability of the gut microbiota, and a lack of agreement across the large number of microbiome studies, this review focused on omics studies that had sampling at more than one time point. A systematic literature search was performed using various combinations of the search terms “Irritable Bowel Syndrome” and “Omics” in the Medline, EMBASE, and Cochrane Library up to 1 December 2022. A total of 16 original studies were reviewed. These multi-omics studies have implicated *Bacteroides*, *Faecalibacterium prausnitzii*, *Ruminococcus* spp., and *Bifidobacteria* in IBS and treatment response, found altered metabolite profiles in serum, faecal, or urinary samples taken from IBS patients compared to the healthy controls, and revealed enrichment in the immune and inflammation-related pathways. They also demonstrated the possible therapeutic mechanisms of diet interventions, for example, synbiotics and low fermentable oligosaccharides, disaccharides, monosaccharides, and polyol (FODMAP) diets on microbial metabolites. However, there was significant heterogeneity among the studies and no uniform characteristics of IBS-related gut microbiota. There is a need to further study these putative mechanisms and also ensure that they can be translated to therapeutic benefits for patients with IBS.

## 1. Introduction

Irritable bowel syndrome (IBS) is the most commonly diagnosed gastrointestinal disorder [1], characterized by recurrent abdominal pain or discomfort and a change in the frequency or form of one’s stools [2]. It is thought to affect around 12% of the global population [1] and is associated with a significant burden of illness as it impacts an individual’s health-related quality of life and work productivity [3].

Despite the prevalence of IBS and the enormous economic disease burden (totalling more than USD 20 billion in the United States alone [4]), the current state of IBS research into the etiopathogenesis and clinical phenotypes of IBS suggest that the condition is heterogenous, multifactorial, and remain incompletely understood [5,6]. There is also no cure or targeted therapy for IBS, and treatments are primarily aimed at providing symptom relief. Burgeoning research and experimental evidence have suggested IBS to be a disorder of the brain–gut–microbiome axis, driven by an altered intestinal and colonic microbiota, abnormal gut immune activation, and increased gut permeability [7]. However, traditional laboratory research is hampered by the lack of a reliable animal model for IBS with poor clinical translation [8], while clinical studies have yielded inconsistent results and may suffer from suboptimal design and power [9].

To overcome these limitations, in recent years, scientific advances in ‘omics’ technologies have enabled precise and accurate molecular measurements (proteins, genes, and metabolite etc.) within a tissue or cell, emerging as a powerful tool to unravel new mechanistic insights beyond the expressed phenotype [10,11]. Omics technologies such as genomics, transcriptomics, proteomics, and metabolomics are increasingly being used to study IBS and improve our understanding of its underlying molecular mechanisms: genomics studies have identified genetic variations associated with IBS and its subtypes. Transcriptomics can provide information on the gene expression changes that occur in response to IBS-related stimuli, while proteomics can identify changes in protein levels and post-translational modifications. Finally, metabolomics is being used to identify changes in the gut microbiota and metabolic pathways that may contribute to IBS symptoms. Alterations in metabolites and metabolite signatures have been associated with central sensitivity pain syndromes including IBS [12]. By integrating the results from different omics platforms, researchers can theoretically gain a more comprehensive view of the complex and multifactorial nature of IBS to inform the development of novel therapeutic strategies for the management of this disorder. 

To the best of our knowledge, there has not been a review focusing on the contributions of longitudinal omics studies to our modern understanding of IBS. This review therefore aimed to systematically synthesise the current body of evidence from omics studies in humans as well as outline possible directions for future omics research and clinical applications. Given the known high inter-individual and day-to-day variability of the gut microbiota due to genetic, diet, environmental, and other factors [13], and a lack of agreement across the large number of faecal microbiome studies [14], this review focused only on studies that had sampling at more than one time point.

## 2. Methods

A systematic literature search was performed in accordance with the Preferred Reporting Items for Systematic Reviews and Meta-Analyses (PRISMA) guidelines [15]. The review protocol was registered under the International Prospective Register of Systematic Reviews (PROSPERO), registration number CRD42022360859.

Various combinations of the key search terms “Irritable Bowel Syndrome” and “Omics” were used in the search strategy for Medline, EMBASE, and Cochrane Library, and searched from database inception through to 1 December 2022. No restrictions on date, language, or subject were implemented on the database search. The detailed search strategy can be found in Appendix A. Abstracts were imported into Microsoft Excel and screened by five independent researchers (C.E.Y., C.Y.L.Y, R.I.H.C., N.Z.-Y.C., and S.E.T.). Full texts were obtained for all abstracts of relevance and their respective reference lists were hand-searched to identify additional relevant articles. Forward searching of prospective citations of the relevant full texts was also performed and authors of the respective articles were contacted if necessary to provide additional data.

Each article was reviewed by at least two researchers blinded to each other’s decision. Disputes were resolved through consensus from the senior author (Q.X.N. or Y.L.L.). The criterion for inclusion were: (1) human studies; (2) utilising omics technology (e.g., proteomics, transcriptomics, genomics, or metabolomics); (3) original published articles; (4) longitudinal study design (with sampling at more than one time point); (5) written or translated into the English language. Animal studies were excluded from this review. A cross-sectional study of the gut microbiome falls short in adequately capturing and reflecting the highly diverse gut ecosystem and dynamic microbiota–gut–brain axis interactions, moreover, the gut microbiome profile tends to vary significantly between individuals from different geographical regions, populations, and even development stages [13]. Studies have also shown that the commonly employed ‘omics’ methods lack accuracy when measuring a single time point [16] and it is more reliable to investigate microbiota changes over time or repeat metabolite analyses; hence, cross-sectional studies were excluded from the present review.

Data from the relevant studies were extracted using a standardized data form in Microsoft Excel including information on the study population, country of origin, type of sample(s) collected and analysed, study methods, time points for taking measurements, and the key study findings. These information were extracted by five independent researchers (C.E.Y., C.Y.L.Y, R.I.H.C., N.Z.-Y.C. and S.E.T.) and cross-checked by a senior author (Q.X.N. or Y.L.L.) for accuracy.

## 3. Results

A total of 16 studies were eligible for inclusion after a systematic literature search (Figure 1). 

Table 1 outlines the key characteristics and findings of the studies reviewed. A total of 16 studies were included in this review [16,17,18,19,20,21,22,23,24,25,26,27,28,29,30,31]. The studies mostly comprised intervention studies that took measurements at the baseline and after intervention (typically four to six weeks) [18,19,21,22,24,25,26,27,28,29,30]. The majority of studies had small sample sizes, with only two having more than 100 patients [20,30]. The studies tended to focus on patients with IBS-D and made use of healthy controls for comparison [16,17,18,20,23,26,27,31], and were generally aimed to identify the microbiota changes and cellular mediators underlying IBS via 16S rRNA gene sequencing, metabolomics, or transcriptomics analyses. 

In terms of the sample type, six studies used stool samples [16,18,19,23,24,25], five used gastrointestinal mucosal samples [16,17,20,25,26], four analysed blood samples [22,27,29,30], and four analysed urine samples [21,27,28,31].

Findings of studies that utilised metagenomics were further elaborated in Table 2. Metagenomics and RNA sequencing are more sensitive, have greater resolution, and provide a more comprehensive picture regarding the structure and function of host microbial communities compared to traditional 16S rRNA sequencing. However, differences in taxa abundances between individuals with IBS and the healthy controls at the baseline and post-intervention appeared rather variable within and inconsistent across the studies (Table 2).

In terms of the metabolomics changes, reduced levels of short-chain fatty acids (SCFAs) have been associated with an altered gut microbiome in IBS [25], while elevated levels of branched-chain amino acids and certain gut peptides have also been observed in IBS patients [23].

## 4. Discussion

Traditional clinical trials involving IBS patients have been confounded by a heterogeneous patient population, highly variable symptoms, and a large placebo effect [9]. IBS consists of a constellation of gut symptoms, and burgeoning research into the gut microbiome has attempted to uncover uniform mechanisms underlying the microbiota–gut–brain axis interactions, especially at the level of metabolite changes and differential gene expressions. The intestinal microbiota comprises billions of diverse bacteria, viruses, fungi, and archaea, and their metabolites and by-products are probably a part of the complex bi-directional microbiota–gut–brain axis [7]. Alterations in the gut microbiome may contribute to the development of IBS symptoms.

As enabled by multi-omics studies, we have some hypotheses on the abnormal alterations in the gut microbiota and microbial metabolites underlying patients with IBS and their symptom flares. Several studies have implicated *Bacteroides*, *Faecalibacterium prausnitzii*, *Ruminococcus* spp., and *Bifidobacteria* in IBS and treatment response [19,24,26]. Similar to the findings of a 2019 review that examined case-control studies detecting gut microbiota in IBS patients [14], increased *Firmicutes* and decreased relative abundance of *Bacteriodetes* were common among faecal microbiota studies, but results for mucosal microbiota were more variable. Metabolomics studies have revealed alterations in the levels of specific metabolites such as SCFAs, bile acids, and amino acids [23,25,27,29], which are the end products of cellular metabolism and can reflect changes in the gut microbiome and other aspects of the gut environment. These changes may contribute to the development of gut symptoms; SCFAs are produced by the gut microbiome and are the primary energy source for intestinal epithelial cells [32]. Reduced levels of SCFAs, particularly butyrate, have been observed in IBS [33] and are thought to reflect alterations in the gut microbiome.

Several studies in this review also found altered metabolite profiles in serum, faecal, or urinary samples taken from IBS patients compared to healthy controls [27,28,29,31], and revealed an enrichment in the immune and inflammation-related pathways [16,22], although the results have been inconsistent. The intestinal mucosa is part of an intricate enteric immune system and comprises a large variety of immune cells [5]; low-grade inflammation and the effects of proinflammatory cytokines and tumour necrosis factor alpha (TNF-alpha) in the colonic mucosa may at least in part explain IBS symptoms and flares [34]. 

These multi-omics studies have also demonstrated the possible therapeutic mechanisms of diet interventions (e.g., synbiotics and low FODMAP diets) on microbial metabolites. It is known that the human gut microbiome can rapidly respond to dietary interventions and an altered diet [35]. Based on the findings of the studies reviewed, oral synbiotic yogurt normalized metabolites are involved in the one-carbon metabolism pathway [27]. Probiotic supplementation increased the counts of presumptive lactic acid bacteria (*Lactobacillus* and *Bifidobacteria*) [19]. Low FODMAP diets resulted in increased 2-hydroxybutyrate in serum and decreased pantothenate in urine [28], while starch- and sucrose-restriction led to increased alpha-linoleic acid and linoleic acid levels in the blood plasma [29], which were likely the direct results of an increased intake of specific foods rich in these essential metabolites. In a double-blind, crossover trial involving paediatric IBS patients compared to non-responders to a low FODMAP diet, responders were characterised to be enriched at the baseline in gut microbes with greater saccharolytic metabolic capacity (family *Bacteroidaceae*, e.g., *Bacteroides*, order *Clostridiales*, e.g., *Ruminococcaceae*, *Dorea,* and *Faecalibacterium prausnitzii*, and family *Erysipilotrichaceae*, e.g., cc_115), while non-responders were enriched at the baseline in *Turicibacter* (from the family *Turicibacteraceae*). [36]. Similarly, in adult IBS patients, non-responders had at the baseline gut dysbiosis, with an overrepresentation of *Streptococcus* and *Dorea* species [37]. This implies that gut microbiota may predict the treatment response. However, the differences in taxa abundances observed at the individual time points can be highly variable and inconsistent when comparing the different time points, and it may not overlap with changes observed in the averaged data [16]. The gut microbiome likely plays a role in the development of IBS symptoms, and patients with certain alterations in the gut microbiome diversity or composition may be more or less likely to respond to particular treatments, however, there are no firm conclusions yet. Metabolomics capture valuable information on metabolites that are either produced endogenously or from the digestion and metabolism of foods. The effects of dietary interventions may be transient and the correlation between the impact on the metabolite signature and long-term symptom control remains unclear. In particular, the effects of probiotic supplementation are likely to be dependent on the baseline host microbiome features and are not sustained [38,39].

To elucidate the effect of microbial metabolism on host function, one study also compared the transcriptional and epigenetic changes [16], hinting at alterations in purine nucleoside phosphorylase and increased purine degradation by gut microbiota in the IBS-D and IBS-C patients. There are complex metabolic changes that occur in IBS and this may have implications for new diagnostic and therapeutic approaches.

The present methods are not without limitations. First, in the majority of studies, 16S rRNA gene sequencing technology was used to detect the faecal microbiota of IBS patients and healthy controls, however, 16S rRNA sequencing provides taxa resolution up to the genus level and is unable to yield information on the functional characteristics compared to newer techniques such as shotgun-metagenome sequencing, which was used in a few studies [16,18,24]. Metagenomics and RNA sequencing are more sensitive, have greater resolution, and provide a more comprehensive picture regarding the structure and function of host microbial communities [40]. Second, at present, the gut microbiome is also primarily studied by the use of stool bacterial communities as a proxy. Stool samples are broadly representative of colonic luminal bacteria; however, some communities of bacteria may be overlooked including those found in the small intestine or embedded within intestinal mucosa [41], which may also explain the difference seen between studies that utilised mucosal as opposed to faecal samples. A study into human gastrointestinal and faecal microbiome found the two to be only partially correlated, and faecal microbiome was a limited indicator of gut mucosa-associated microbiome composition and metagenomic function [38]. As different parts of the intestinal tract contain different luminal and mucosal commensal microbiota, we should collect gut microbiota at different sites. Last but not least, although this review focused on the time-series feature of available studies, most studies only took measurements at two time points, and it is necessary to have more studies with greater longitudinal sampling to reliably investigate microbiota and metabolite changes over time. Longitudinal sampling would be particularly useful to compare different disease states such as IBS flare versus remission or the effects of interventions. Although the gut microbiota is a potential biomarker for IBS, there is no firm conclusion on the characteristics of IBS-related gut microbiota, and no biomarkers have been identified to date.

## 5. Conclusions

In conclusion, having reviewed a range of data types and reported pathways that were identified across the studies, there was significant heterogeneity among the studies and no uniform characteristics of IBS-related gut microbiota. There is still a paucity of human studies and a need to ensure that these putative mechanisms can be translated to therapeutic benefits for patients with IBS. Despite the advances in metabolomics and microbiome understanding, studies have not uncovered distinct changes that underlie the symptoms in IBS patients.

## Figures and Tables

**Figure 1 metabolites-13-00484-f001:**
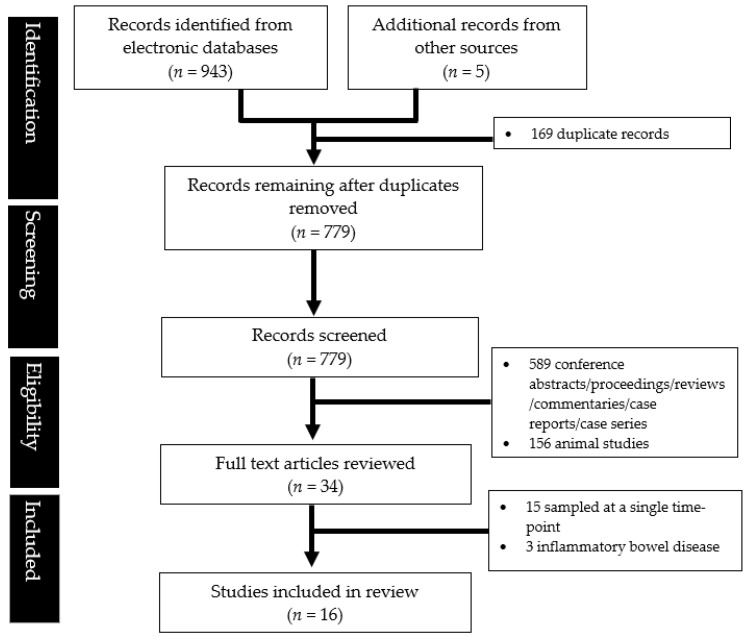
PRISMA flowchart showing the study abstraction process.

**Table 1 metabolites-13-00484-t001:** The characteristics and key findings of the studies reviewed (arranged alphabetically by the first author’s last name).

Study, Year	Country	Study Population	Test Sample	Study Methods	Study Time points	Key Findings
Aerssens, 2008 [17]	United States	36 IBS patients (21 IBS-D and 15 IBS-C) and 25 healthy controls; Rome II criteria for diagnosis	Colon biopsy samples	Histological assessment, microarray analysis and real-time quantitative PCR	Two time points, ~3 months interval	- differentially expressed genes suggest functional alterations of host mucosal immune response to microbial pathogens - compared to healthy controls, IBS patients had significantly increased expression of DKFZP564O0823 (an uncharacterized gene)
Ankersen, 2021 [18]	Denmark	34 IBS patients (either IBS-D or IBS-M) randomised to receive either web-based low FODMAP diet intervention or probiotic; Rome III criteria for diagnosis	Stool sample	CeGaT GmbH culture-independent, whole-genome, shotgun metagenomic, next-generation sequencing of genomic DNA	Four time points, during the course of 1 year	- web-based low FODMAP diet intervention and probiotic intervention were both efficacious in managing IBS symptoms - in both cases, treatment response could not be predicted or explained by the composition of host gut microbiota
Bonfrate, 2020 [19]	Italy	25 IBS patients randomised to receive either probiotic or placebo; Rome IV criteria for diagnosis	Stool sample	Counts of viable bacteria and community level catabolic profiles; chromatogram peak identification for faecal metabolome	Four time points, 0, 30, 45 and 60 days	- probiotic supplementation increased counts of presumptive lactic acid bacteria (*Lactobacillus* and *Bifidobacteria*) - relative abundance of propanoic, butanoic, pentanoic acids and hydrocarbons, but decreased phenol post-probiotic supplementation
Ek, 2015 [20]	Sweden	534 IBS patients and 4932 healthy controls; adapted version of Rome II criteria	GWA genotyping data and rectal mucosal biopsies	Genotyping with Illumina OmniExpress arrays, SNP-expression quantitative trait loci interactions testing; real-time PCR for candidate gene expression	GWA study in a general population sample, followed by case-control cohorts to study suggestive association signals	- suggestive locus on chromosome 7p22.1, which showed genetic risk effects replicated in all case-control cohorts albeit did not achieve genome-wide significance
Kim, 2019 [21]	South Korea	63 IBS-D patients randomised to receive either probiotics or placebo; Rome II criteria	Urine sample	Q-TOF-MS metabolomics	Two time points, baseline and after 8 weeks	- urinary levels of palmitic acid methyl ester (PAME), cholic acid, and palmitoleoyl ethanolamide (PEA) were increased with probiotic intervention - among responders, urinary levels of PAME showed significant correlation with improvements in IBS severity scoring system (IBS-SSS) scores
Kuo, 2015 [22]	United States	19 IBS patients and 29 IBD patients enrolled in a 9-week relaxation response based mind-body group intervention; Rome III criteria	Whole blood sample	Total RNA isolated from whole blood samples, high-throughput Affymetrix GeneTitan system peripheral blood transcriptome	Two time points, at baseline and after 9 weeks	- significant differences in the expression profile of genes related to inflammation (such as nuclear factor-κB (NF-κB)) and kinases (such as ERK1/2, MAPK8, MAPK and MAP3K7) in both IBS and IBD patients post-intervention
Le Gall, 2011 [23]	United Kingdom	10 IBS patients, 13 ulcerative colitis patients and 22 healthy controls; Rome III criteria	Stool sample	High resolution 1H NMR-based metabolomics	Four time points, over 2 years	- NMR spectra for IBS samples were variable, with poor differentiation between IBS patients and healthy controls as compared to UC patients and healthy controls - decreased relative abundance of branched chain fatty acids (BCFA) in IBS relative to controls, which may be the result of reduced number of BCFA producing bacteria
Mack, 2020 [24]	Germany	22 IBS patients subjected to two weeks of sugar elimination diet and 7 IBS patients used as controls; Rome IV criteria	Stool sample	16S rRNA amplicon and shotgun-metagenome sequencing	Three time points, at baseline, after 2-weeks sugar elimination diet and after 4-weeks tolerance phase	- Alpha and beta diversity of 16s rRNA-based faecal microbiota composition did not differ much between responders and non-responders to diet intervention - however, shotgun-metagenomics showed significant differences in pathways encoding starch degradation and complex amino acid biosynthesis, with involvement of *Faecalibacterium prausnitzii*, *Ruminococcus* spp. and *Bifidobacterium longum*
Mars, 2020 [16]	United States	29 IBS-D, 22 IBS-C and 24 healthy controls; Rome III criteria	Mucosal biopsy and stool samples	16S rRNA sequencing and metagenome sequencing	>1 time point, baseline and then monthly for 6 months	- immune and inflammation-related pathways were enriched among IBS-D and IBS-C patients compared to healthy subjects - increased purine breakdown by gut microbiota in IBS patients - *Halobiforma nitratireducens*, an Archaea, was consistently elevated in the flare samples from IBS-D and IBS-C patients
Moser, 2019 [25]	Austria	10 IBS-D patients treated with an oral synbiotic for four weeks; S3 guidelines for diagnosis	Gastrointestinal mucosal and stool samples	Fluorescence activated cell sorting analysis and 16S rRNA gene analysis of gastrointestinal mucosal specimens; GC-MS analysis and 16S rRNA gene analysis of stool samples	Two time points, baseline and after 4 weeks	- after four weeks of oral synbiotic, there was increased microbial diversity in gastric and duodenal mucosal samples - SCFAs and butyrate were elevated in faecal samples while faecal zonulin concentration was decreased
Ng, 2013 [26]	Hong Kong	10 IBS patients and 10 healthy controls treated with oral probiotic mix VSL#3 for four weeks; Rome III criteria	Rectal biopsy samples	16S rRNA gene sequencing	Two time points, baseline and after 4 weeks	- at baseline, IBS patients had lower gut microbiota diversity and evenness than controls, with increased relative abundance of *Bacteroidetes* and *Synegitestes*, and reduced abundance of *Actinobacteria* - oral probiotic supplementation reduced *Bacteroides* in IBS patients to levels comparable to healthy controls
Noorbakhsh, 2019 [27]	Iran	8 IBS-D patients and 16 healthy controls given synbiotic yoghurt for four weeks; Rome III criteria	Urine and serum specimens	1H NMR-based metabolomics	Two time points, baseline and after 4 weeks	- serum and urine metabolite concentrations were significantly different at baseline, between IBS-D patients and healthy controls - synbiotic yoghurt increased serum branched-chain amino acids and induced a shift in one-carbon metabolism - faecal *Lactobacilli* increased after four weeks of intervention
Nybacka, 2021 [28]	Sweden	56 IBS patients randomised to low FODMAP or traditional diet interventions for four weeks; Rome III criteria	Urine and serum specimens	1H NMR-based metabolomics	Two time points, baseline and after 4 weeks	- no distinct clustering patterns or trends in serum and urine metabolites at baseline - responders to low FODMAP diet had increased 2-hydroxybutyrate in serum and decreased pantothenate in urine metabolites, which could reflect changes in dietary consumption of foods
Stenlund, 2021 [29]	Sweden	91 IBS patients randomised to starch and sucrose restricted diet or control for four weeks; Rome IV criteria	Blood plasma specimen	GC-MS and LC-MS based metabolomics	Two time points, baseline and after 4 weeks	- enriched linoleic acid metabolism, fatty acid biosynthesis, and beta-oxidation in the intervention group - increased alpha-linoleic acid and linoleic acid levels in blood plasma post diet intervention, likely due to changes in dietary consumption of foods
Wang, 2022 [30]	United States	188 IBS patients randomised to placebo treatment for six weeks	Blood sample	Genotyping with Illumina (Infinium Global Screening Array v2.0) and RNA sequencing	Two time points, baseline and after 6 weeks	- IBS patients who are homozygous for rs4680 met (met/met) had the greatest placebo response - molecular mechanisms related to EGR1 gene expression appear common in varying forms of placebo response, even among IBS patients
Yamamoto, 2019 [31]	Canada	42 IBS patients and 20 healthy controls; Rome III criteria	Urine specimen	Q-TOF-MS based nontargeted metabolomics	Two time points, 6 weeks apart	- ten urinary metabolites (two glycosylated hydroxylysine metabolites, a glycated tryptophan analogue, a modified nucleoside, amino acids lysine, serine, ornithine, and glutamine and other amino acid catabolites, dimethylglycine, and imidazole propionate) were consistently elevated in IBS patients as compared to healthy controls

Abbreviations: FODMAP, fermentable oligosaccharides, disaccharides, monosaccharides and polyols; GC-MS, gas chromatography-mass spectrometry; GS-FLX, Genome Sequencer FLX System; IBS, irritable bowel syndrome; IBS-D, IBS-diarrhoea; IBS-M, IBS-mixed type; NMR, nuclear magnetic resonance; PCR, polymerase chain reaction; QIIME, quantitative insights into microbial ecology; SCFA, short-chain fatty acids; SNP, single nucleotide polymorphism; Q-TOF-MS, quadrupole time-of-flight mass spectrometry.

**Table 2 metabolites-13-00484-t002:** Comparisons in terms of taxa for metagenomics studies.

Study, Year	Taxonomy of Microbiota		Significant Alterations
Ankersen, 2021 [18]	Low FODMAP diet group *Streptococcacae* (*family*), *Streptococcus sp001556435* (*species*); *Ruminococcaceae* (*family*), *MGYG-HGUT-03337* (*species*); *Ruminococcaceae* (*family*), *MGYG-HGUT-02040* (*species*); *Ruminococcaeceae* (*family*), *Faecalibacterium prausnitzii_H* (*species*); *Peptostreptococcaceae* (*family*), *Romboutsia timonensis* (*species*; *Peptococcaceae* (*family*), *MGYG-HGUT-04093* (*species*); *Oscillospiraceae* (*family*), *MGYG-HGUT-02704* (*species*); *Oscillospiraceae* (*family*), *MGYG-HGUT-02673* (*species*); *Oscillospiraceae* (*family*), *MGYG-HGUT-02143* (*species*); *Oscillospiraceae* (*family*), *MGYG-HGUT-00703* (*species*); *Oscillospiraceae* (*family*), *Flavonifractor plautii* (*species*); *Oscillospiraceae* (*family*), *ER4 sp003522105* (*species*); *Lachnospiraeceae* (*family*), *TF01-11 sp000436755* (*species*); *Lachnospiraeceae* (*family*), *MGYG-HGUT-01758* (*species*); *Lachnospiraeceae* (*family*), *MGYG-HGUT-01052* (*species*); *Lachnospiraeceae* (*family*), *Lachnospira eligens_B* (*species*); *Lachnospiraeceae* (*family*), *CAG-95 sp900066375* (*species*); *Butyricicoccaceae* (*family*), *MGYG-HGUT-01115* (*species*); *Bacteroidaceae* (*family*), *Bacteroides caccae* (*species*)	Probiotic group *Streptococcaeceae* (*family*), *Streptococcus thermophilus* (*species*); *Ruminococcaeceae* (*family*), *Faecalibacterium prausnitzii_H* (*species*); *Lactobacillaceae* (*family*), *Lactobacillus_F plantarum* (*species*); *Lactobacillaceae* (*family*), *Lactobacillus_C paracasei* (*species*); *Lactobacillaceae* (*family*), *Lactobacillus adiophilus* (*species*); *Lachnospiraceae* (*family*), *MGYG-HGUT-04609* (*species*); *Lachnospiraceae* (*family*), *Acetatifactor sp900066365* (*species*); *Bifidobacteriaceae* (*family*), *Bifidobacterium animalis* (*species*); *Bacteriodaceae* (*family*), *Bacteroides eggerthii* (*species*); *Acutalibacteraceae* (*family*), *Clostridium_A leptum* (*species*)	- in the low FODMAP group, compared to the baseline, Streptococcacae, Streptococcus sp001556435; Ruminococcaceae, MGYG-HGUT-03337; Ruminococcaceae, MGYG-HGUT-02040; Peptococcaceae, MGYG-HGUT-04093; Oscillospiraceae, Flavonifractor plautii; Oscillospiraceae, Flavonifractor plautii; Oscillospiraceae, ER4 sp003522105; Lachnospiraeceae, MGYG-HGUT-01758; Lachnospiraeceae, CAG-95 sp900066375 were significantly altered (*p* < 0.05) - in the probiotic group, compared to the baseline, Streptococcaeceae, Streptococcus thermophilus; Bacteroides eggerthii; Acutalibacteraceae, Clostridium_A leptum were significantly altered (*p* < 0.05)
Mack, 2020 [24]	Adlercreutzia (genera); Ruminococcus (genera); *Coriobacteriaceae* (*family*); *Christensenellaceae* (*family*); *Ruminococcaceae* (*family*)		- at the baseline, five taxa were significantly different between IBS patients with sugar malabsorption and the healthy controls (*p* < 0.05)
Mars, 2020 [16]	*Clostridium innocuum* (*species*); *Trueperella pyogenes* (*species*); *Citrobacter freundii* (*species*); *Ruminococcus sp. AT 10* (*species*); *Granulicatella elegans* (*species*); *Streptococcus caballi* (*species*); *Streptococcus intermedius* (*species*); *Ruminococcus torques* (*species*); *Enterobacter lignolyticus* (*species*); *Streptococcus oralis* (*species*); *Streptococcus gordonii* (*species*); *Streptococcus mutans* (*species*); *Streptococcus pneumoniae* (*species*); *Streptococcus parasanguinis* (*species*); *Eubacterium brachy* (*species*); *Prevotella baroniae* (*species*); *Gardnerella vaginalis* (*species*)		- two bacterial families had higher relative abundances in IBS compared to healthy controls (*p* < 0.05) - four *Streptococcus* species had higher relative abundances in IBS compared to the healthy controls (*p* < 0.05)

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
