# Peer review of "What Has Longitudinal ‘Omics’ Studies Taught Us about Irritable Bowel Syndrome? A Systematic Review"

_metabolites, 2023, doi:10.3390/metabo13040484_

Round 1

Reviewer 1 Report (Previous Reviewer 1)

This revised manuscript can be accepted. No further comments.

Author Response

Thank you for the kind words.

Reviewer 2 Report (Previous Reviewer 3)

The authors answered my questions. I do not have other concerns. But there is still one error that they did not fix:

1.     Lines 128-129: “six studies used stool samples [16,18,19,23,24,25] five used…” -> “six studies used stool samples [16,18,19,23,24,25], five used…”. Please add the comma in front of the word “five”.

Author Response

Thank you for the comment. We have added the comma before the word five.

Reviewer 3 Report (New Reviewer)

This nice review did a systematic and comprehensive literature search on omic studies of gut microbes and also host responses in IBS patients. The authors did well in cataloging and describing their approach that ended with 16 studies that fulfilled their defined criteria. The results are well outlined in the two large tables, and the discussion points to future directions in this area. I enjoyed reading this paper and it can be published as is.

Author Response

Thank you for taking the time to review the paper and for your kind words and comments.

This manuscript is a resubmission of an earlier submission. The following is a list of the peer review reports and author responses from that submission.

Round 1

Reviewer 1 Report

The paper by Dr. Qin Xiang Ng describes longitudinal omics studies contribute to investigate mechanisms underlying patients with Irritable Bowel Syndrome (IBS) and their symptom flares whereas their results were still too poor to uncover fully understanding of IBS. IBS has been associated with mucosal dysfunction, mild inflammation, and altered colonic bacteria therefore omics have been used to evaluate the alteration. The authors listed several omics studies to synthesize the current body of evidence however the table was complicated. It included several different type of samples and analysis. Furthermore, it is unclear why the author focus on only omics studies and what this review contributes in this field. According to the reasons, the manuscript can not be accepted for publication in Metabolites.

Author Response

  1. Thank you for the comments. We have reorganized our table of results to emphasize the time-series feature for each paper and trimmed away certain unnecessary information.
  2. In our introduction we have tried to more clearly outline the rationale for the present study. Given the lack of understanding about the putative mechanisms of IBS, we have chosen to focus on the findings of omics technologies, such as genomics, transcriptomics, proteomics, and metabolomics, which provide rich information about underlying molecular mechanisms. Genomics studies have identified genetic variations associated with IBS and its subtypes. Transcriptomics can provide information on the gene expression changes that occur in response to IBS-related stimuli, while proteomics can identify changes in protein levels and post-translational modifications. Metabolomics is being used to identify changes in the gut microbiota and metabolic pathways that may contribute to IBS symptoms. Alterations in metabolites and metabolite signatures have been associated with central sensitivity pain syndromes, including IBS. Hence the current focus on omics.

Reviewer 2 Report

The review submitted  by Xiang et al. is focused on longitudinal ‘omics’ studies to characterize irritable bowel syndrome (IBS). The idea of exploring the value of longitudinal studies in this context is interesting but the results should be expanded and presented more effectively, to be really useful for researchers in the field. The manuscript must be revised, before being suitable for publication in Metabolites.

The main flaw of this review is the Results section. The Results section essentially contains only Table 1 with some characteristics and the key findings of the 17 reviewed manuscripts. Compared to a classic review the number of documents cited is very small, I would therefore expect a greater depth and an analysis of what has been found in common in several articles and what has not, perhaps with adequate graphical supports. A precise analysis should be added in this section, possibly grouping the screened articles in convenient ways (e.g. grouping by sample type), in order to add useful information for the readers (e.g. for metagenomics studies a comparison in terms of taxa might be relevant). 

As for the form of Table 1, I suggest to add a separate column for the ‘Sample’ (feces, urine, serum, plasma, …), leaving in the column ‘Methods’ only the applied methods.  Does the column ‘Country’ refer to the cohort of patients analyzed within the study? In this case, it may be relevant but it should be stated somewhere. Since all the considered studies are longitudinal, time points should also be added in a separate column. 

As concerns the search strategy in the supplementary material, a brief clear description of the three procedures used on the three websites with an indication of the number of documents found in each search could be useful to replicate the analysis. In fact, visiting the websites of OVID medline, EMBASE and Cochran library is not so intuitive to perform a search and, when trying, the results are not the same as presented. 

The protocol CRD42022360859 deposited in PROSPERO is generic and does not contain details on the procedure used and the criteria for selecting articles for the review. In the protocol, it only says that ‘Include studies that conduct any omic analysis on IBS patients’, nowhere it is specified that only longitudinal studies are considered. The protocol deposited must be consistent with what is written in the Methods section of the review.  Moreover, it is not clear whether one of the search was performed in OVID medline (as stated in the review) or in MEDLINE (as stated in the deposited protocol).

Author Response

Thank you for the rich comments and suggestions!

  1. We have reorganized Table 1 by trimming away some excess information and adding two new columns, one to emphasize the time-series feature of each study and another to state the sample type, as suggested. Yes, the cohort does relate to the country of origin for the sample of patients recruited and analysed.
  2. We have also added more text descriptions in the results section, "Table 1 outlined the key characteristics and findings of the studies reviewed. The studies comprised mostly of intervention studies that took measurements at baseline and after intervention (typically four to six weeks) [18,19,21,22,24-30]. Majority of the studies had small sample sizes, with only two having more than 100 patients [20,30]. The studies tended to focus on patients with IBS-D and made use of healthy controls for comparison [16,17,18,20,23,26,27,31], and generally aimed to identify the microbiota changes and cellular mediators underlying IBS via 16S rRNA gene sequencing, metab-olomics or transcriptomics analyses. In terms of the sample type, six studies used stool samples [16,18,19,23,24,25] five made use of gastrointestinal mucosal samples [16,17,20,25,26], five analysed blood samples [22,27,29,30] and four analysed urine samples [21,27,28,31]. Differences in taxa abundances between individuals with IBS and healthy controls at baseline and post-intervention appeared rather variable within and inconsistent across the studies. In terms of metabolomics changes, reduced levels of short-chain fatty acids (SCFAs) have been associated with an altered gut microbiome in IBS [25], while elevated levels of branched-chain amino acids and certain gut peptides have also been observed in IBS patients [23]."
  3. As advised, we have added the number of results found for each platform in our Supplementary Material.
  4. Apologies for the confusion, we used MEDLINE and not OVID MEDLINE as stated in the manuscript. We have now corrected this.
  5. We have submitted an amendment to our original PROSPERO protocol to state that we only include RCTs or cohort studies and also specified the additional inclusion criteria, i.e. only studies that had sampling at more than one time-point.

Reviewer 3 Report

Ng et al. attempted to summarize time-series multi-omics studies of IBS (Irritable Bowel Syndromes) published in the past. To achieve this, they did a rather exhaustive search of ‘Omics’ and ‘IBS’ on OVID Medline, EMBASE, and Cochrane Library and filtered out some literature that did not satisfy their requirements. Eventually, they found 17 papers and summarized their achievements in a table. It is great that the authors spent the time to summarize relevant research, but I believe the presentation can be improved. I hope my comments will help them to improve the readability of the paper. I am quite open to looking at a revised version if the authors could address some issues in a satisfactory fashion, which we describe in more detail below.

1.     My major concern is the lack of a text description in the results section. I am looking forward to a more detailed description of the commonalities and differences among found papers.

2.     The authors highlighted the importance of time series (i.e., multiple measurements at different times). However, in Table 1, they did not clearly capture the number of measurements at different times and how many days/weeks/months apart from each other. I would recommend authors add a column to summarize the time-series feature for each paper.

3.     Some acronyms occurred without their full names being shown, so some readers might not understand their meanings. For example, FODMAP appeared on line 27 for the first time without the full name. The full name “fermentable oligosaccharides, disaccharides, monosaccharides, and polyols” shall be added.

4.     I would appreciate it if the authors could give an outlook for future directions based on their summarized results here. For instance, do we need more time-series measurements or large sample sizes?

Author Response

Thank you for the comments and suggestions!

  1. We have added more text descriptions in the results section, "Table 1 outlined the key characteristics and findings of the studies reviewed. The studies comprised mostly of intervention studies that took measurements at baseline and after intervention (typically four to six weeks) [18,19,21,22,24-30]. Majority of the studies had small sample sizes, with only two having more than 100 patients [20,30]. The studies tended to focus on patients with IBS-D and made use of healthy controls for comparison [16,17,18,20,23,26,27,31], and generally aimed to identify the microbiota changes and cellular mediators underlying IBS via 16S rRNA gene sequencing, metab-olomics or transcriptomics analyses. In terms of the sample type, six studies used stool samples [16,18,19,23,24,25] five made use of gastrointestinal mucosal samples [16,17,20,25,26], five analysed blood samples [22,27,29,30] and four analysed urine samples [21,27,28,31]. Differences in taxa abundances between individuals with IBS and healthy controls at baseline and post-intervention appeared rather variable within and inconsistent across the studies. In terms of metabolomics changes, reduced levels of short-chain fatty acids (SCFAs) have been associated with an altered gut microbiome in IBS [25], while elevated levels of branched-chain amino acids and certain gut peptides have also been observed in IBS patients [23]."
  2. We thank the reviewer for the suggestion and have added a column to Table 1 to emphasize the time-series feature of each study.
  3. Apologies for the oversight. We have now defined this abbreviation in the first instance of its use. We have also added an explanation of the abbreviations used in Table 1 below the table.
  4. Given that omics studies are costly to conduct and difficult to analyse (given the large volume of data it generates), both approaches have its own set of challenges. Nonetheless, we would suggest that we need more studies with greater longitudinal sampling to reliably investigate microbiota and metabolites changes over time. As added in our discussion section, "Longitudinal sampling would be particularly useful for comparing different disease states like IBS flare versus remission or the effects of interventions." Although the gut microbiota is a potential biomarker for IBS, there is no firm conclusion on the characteristics of IBS-related gut microbiota, and no biomarkers have been identified to date.

Round 2

Reviewer 1 Report

The authors have made substantial revisions in response to the original critique.

Reviewer 2 Report

The review submitted by Xiang et al. is focused on longitudinal ‘omics’ studies to characterize irritable bowel syndrome (IBS). The revised manuscript is still not suitable for publication in Metabolites. 

The main flaw of this review is the Results section. Although Table 1 has been improved following the comments of the first revision round, the Authors added only one sentence in the Results section. The whole Methods section essentially still contains only Table 1 with some characteristics and the key findings of the reviewed manuscripts. Moreover, the number of the manuscripts considered by the Authors in the section Methods dropped from 17 to 16, without any mention from them in their response to reviewers. 

Compared to a classic review the number of documents cited is very small, I would therefore expect a greater depth and an analysis of what has been found in common in several articles and what has not, perhaps with adequate graphical supports. A precise analysis should be added in this section, possibly grouping the screened articles in convenient ways (e.g. grouping by sample type), in order to add useful information for the readers (e.g. for metagenomics studies a comparison in terms of taxa might be relevant). 

A brief clear description of the three procedures used on the three websites, useful to replicate the analysis, has not been added in the supplementary material. 

The protocol CRD42022360859 deposited in PROSPERO is still generic and does not contain details on the procedure used and the criteria for selecting articles for the review. In the protocol, it only says that ‘Include studies that conduct any omic analysis on IBS patients’, nowhere it is specified that only longitudinal studies are considered. The protocol deposited must be consistent with what is written in the Methods section of the review. 

The Authors stated that they “have submitted an amendment to their original PROSPERO protocol”. If so, they should have waited until the record in PROSPERO had been corrected before resubmitting their work for revision.

Reviewer 3 Report

The authors answered my questions. But I detected some grammatical errors that need to be solved:

1.     Line 114: “Table 1 outlined…” -> “Table 1 outlines…

2.     Line 116: “Majority of the studies…” -> “The majority of the studies…”

3.     Line 122: “six studies used stool samples [16,18,19,23,24,25] five…” -> “six studies used stool samples [16,18,19,23,24,25], five…”